# *Mitragyna* Species as Pharmacological Agents: From Abuse to Promising Pharmaceutical Products

**DOI:** 10.3390/life12020193

**Published:** 2022-01-27

**Authors:** Islamudin Ahmad, Wisnu Cahyo Prabowo, Muhammad Arifuddin, Jaka Fadraersada, Niken Indriyanti, Herman Herman, Reza Yuridian Purwoko, Firzan Nainu, Anton Rahmadi, Swandari Paramita, Hadi Kuncoro, Nur Mita, Angga Cipta Narsa, Fajar Prasetya, Arsyik Ibrahim, Laode Rijai, Gemini Alam, Abdul Mun’im, Sukanya Dej-adisai

**Affiliations:** 1Department of Pharmaceutical Sciences, Faculty of Pharmacy, Universitas Mulawarman, Samarinda 75119, Indonesia; marifuddin@farmasi.unmul.ac.id (M.A.); jakafadraersada@gmail.com (J.F.); niken.indriyanti@gmail.com (N.I.); mita@farmasi.unmul.ac.id (N.M.); 2Pharmaceutical Research and Development Laboratory of FARMAKA TROPIS, Faculty of Pharmacy, Universitas Mulawarman, Samarinda 75119, Indonesia; wisnu@farmasi.unmul.ac.id (W.C.P.); herman.mulawarman@gmail.com (H.H.); kuncoro_hadi82@yahoo.com (H.K.); anggaciptan@yahoo.com (A.C.N.); fajarprasetya@farmasi.unmul.ac.id (F.P.); achie.ibrahim@gmail.com (A.I.); najwankhanrjai@yahoo.co.id (L.R.); 3Faculty of Military Medicine, Universitas Pertahanan RI, Bogor 16810, Indonesia; reza.purwoko@idu.ac.id; 4Faculty of Pharmacy, Hasanuddin University, Makassar 90245, Indonesia; firzannainu@unhas.ac.id (F.N.); daengta007@yahoo.com (G.A.); 5Department of Agricultural Product Technology, Faculty of Agriculture, Universitas Mulawarman, Samarinda 75119, Indonesia; arahmadi@unmul.ac.id; 6Research Center of Natural Products from Tropical Rainforest (PUI-PT OKTAL), Department of Community Medicine, Faculty of Medicine, Universitas Mulawarman, Samarinda 75119, Indonesia; s.paramita@fk.unmul.ac.id; 7Laboratory of Pharmacognosy-Phytochemistry, Faculty of Pharmacy, Universitas Indonesia, Depok 16424, Indonesia; munim@farmasi.ui.ac.id; 8Department of Pharmacognosy and Pharmaceutical Botany, Faculty of Pharmaceutical Sciences, Prince of Songkla University, Hat-Yai, Songkhla 90110, Thailand; sukanya.d@psu.ac.th

**Keywords:** 7-hydroxymitragynine, kratom, *Mitragyna speciosa*, mitragynine

## Abstract

*Mitragyna* is a genus belonging to the Rubiaceae family and is a plant endemic to Asia and Africa. Traditionally, the plants of this genus were used by local people to treat some diseases from generation to generation. *Mitragyna speciosa* (Korth.) Havil. is a controversial plant from this genus, known under the trading name “kratom”, and contains more than 40 different types of alkaloids. Mitragynine and 7-hydroxymitragynine have agonist morphine-like effects on opioid receptors. Globally, *Mitragyna* plants have high economic value. However, regulations regarding the circulation and use of these commodities vary in several countries around the world. This review article aims to comprehensively examine *Mitragyna* plants (mainly *M. speciosa*) as potential pharmacological agents by looking at various aspects of the plants. A literature search was performed and information collected using electronic databases including Scopus, ScienceDirect, PubMed, directory open access journal (DOAJ), and Google Scholar in early 2020 to mid-2021. This narrative review highlights some aspects of this genus, including historical background and botanical origins, habitat, cultivation, its use in traditional medicine, phytochemistry, pharmacology and toxicity, abuse and addiction, legal issues, and the potential of *Mitragyna* species as pharmaceutical products.

## 1. Introduction

*Mitragyna* is a genus belonging to the Rubiaceae family, also known as the coffee family [1]. There are ten species in this genus. As many as 0% of the species of this genus are native plants and are widely found in various regions in Asia, including India, Bangladesh, Malaysia, Thailand, Vietnam, the Philippines, New Guinea, and Indonesia (Sumatra and Kalimantan) [1,2]. The remaining species are dispersed across mainland Africa and East Asia [3,4]. Plants of this genus grow in swampy areas, along riverbanks, and in areas often inundated by water [3]. Traditionally, the *Mitragyna* genus has been trusted and used by the local community to treat various diseases such as fevers, malaria, diarrhea, coughs, hypertension, diabetes mellitus, muscle pain, and worm infections [5]. Scientifically, it has also been reported that *Mitragyna* has a pharmacological effect that has the potential to be a source of raw materials for drugs that have the following effects: anti-inflammatory and antinociceptive [6,7]; anti-obesity [8]; analgesic [9,10,11,12]; antipyretic, sedative, stimulant, and depressant [13,14,15,16]; antidopaminergic [17]; effects on memory [18]; antidiarrheal [19,20]; antioxidant [21,22]; and antimicrobial [21].

*Mitragyna* plants contain more than 40 types of alkaloids [23], especially mitragynine (66.2%) and its derivatives, paynantheine (8.6%), speciogynine (6.6%), 7-hydroxymitragynine (2%), and speciociliatine (0.8%) [24], which have an agonist *morphine-like* effect on opioid receptors [25,26]. *Mitragyna speciosa* (Korth.) Havil. is the species most commonly reported to contain psychoactive alkaloids [27]. It is native to East Kalimantan, Indonesia, where it is known by the local name “kadamba,” known in the world market as “kratom.” This plant is mostly grown and cultivated in Indonesia’s Kalimantan and is exported to European and American countries.

The available products of this plant are traded online in liquid, powder, and extract forms [27,28]. *M. speciosa* has a high economic value, especially in dried leaf powder and extract forms. For 5 g of kratom extract, the price starts at $35 to $45 [24]. Kratom is illegal in some countries, although it has been legalized in other countries [29,30,31]. Many publications [32,33,34,35,36,37,38] have reported the abuse and addictive effects of this plant. The Drug Enforcement Administration (DEA) of the United States has added this plant to its list of drugs of concern, and the Food and Drug Administration (FDA) has issued a press release identifying it as an opioid with the potential for abuse, although the literature on its therapeutic and adverse effects is still lacking [39]. On the other hand, this plant also has the potential to be a source of medicinal raw materials that can be processed into profitable pharmaceutical products.

This review article aims to comprehensively examine *Mitragyna* plants (mainly *M. speciosa*) as potential pharmacological agents through the investigation of aspects such as historical background and botanical origins, habitat, cultivation, the application of *Mitragyna* in traditional medicine and ethnomedical uses, phytochemistry, pharmacology, and toxicity, abuse and addiction, legal issues, and the potential of *Mitragyna* plants as pharmaceutical products.

## 2. Research Methodology

In the current narrative review study, the literature search was performed from March 2020 to July 2021 using electronic databases such as PubMed, Embase, Scopus, Web of Science, Directory of Open Access Journals (DOAJ), National Health Institute (NIH), and Google Scholar. The relevant search terms for all aspects of *Mitragyna* plants were (*Mitragyna* Korth OR *Mitragyna* Genus OR *Mitragyna* OR kratom OR mitragynine) AND (botanical OR ethnobotany OR morphology OR taxonomy OR habitat OR cultivation) AND (traditional medicine OR ethnomedical uses) AND (pharmacology OR pharmacokinetic OR toxicology of *Mitragyna*) AND (phytochemistry OR phytoconstituent). A literature search for relevant abuse, addiction effects used the following keywords: (*Mitragyna* OR kratom OR mitragynine) AND (abuse or addiction or human effect) AND (legal status).

## 3. Historical Background and Botanical Origins

*Mitragyna* is a paleotropic genus of medium size that is widely found mainly in tropical and subtropical areas and is native to Africa [2] and Asia [40]. The genus *Mitragyna* was first described by the Dutch botanist Pieter Willem Korthlas (1807–1892), who discovered the plants and created significant botanical collections in the Malay Archipelago. Historically, this genus has gone through various studies and processes of compiling and placing sub-families into various families, finally leading to Robbrecht re-entering this genus into the *Rubiaceae* family in 1993 [4]. The Rubiaceae family, commonly called the coffee or madder family, is one of the most prominent plant families [41]. Rubiaceae can be classified into several subfamilies, such as Cinchonoideae, Ixoroideae, and Rubioideae, with Cinchonoideae being the smallest member (~120 genera, including *Mitragyna*) [42]. This family is known for features such as round petal flowers, inferior ovaries, interpetiolar spots, and simple crossed or slightly circular leaves [43,44].

*Mitragyna* is one of the rubiaceous genera (Rubiaceae) that were once grouped with the Naucleaceae family, with dense, round, head-like inflorescences [45,46]. Plants of the genus *Mitragyna* are shrubs or trees characterized by a dense inflorescence system. The flowers are arranged in compact round heads with pale interflora spathuloid bracts. The flowers and ovaries are fused, and the fruit is abundant [44]. They have two-celled ovaries with a cylindrical stigma, opposite leaves, and keel-shaped stalks. The flower’s head lies on the side shoots laterally, forming a short or long tube with five lobes. The flowers have heart-shaped stamens with heads containing lanceolate anthers [43,45,46,47]. Plants of the genus *Mitragyna* consist of ten species (six species are found in Asia to Southeast Asia, and the four other species are found in Africa), as shown in Table 1.

*M. speciosa* is best known as “kratom” but is also known as giam in Vietnam, thom, ithang, gratom, bai krathom, kakaum, and kratom in Thailand, kadamba, purik, and keton in Indonesia, ketum, sepat, biak-biak, and kutum in Malaysia, mambog, lugug, and polapupot in the Philippines, and beinsa, bein-sa-ywat in Myanmar. Information on this species was first published by Pieter Willem Korthals (a botanist from the Netherlands, 1807–1892) under the name *Mitragyna*. However, the proposal was considered invalid because it was not accompanied by the required botanical descriptions according to the International Code of Botanical Nomenclature. Similar to other genera, some experts also renamed *Mitragyna* plants by grouping them into the genus *Naucleae*, under names such as *N. korthalsii*, *N. luzonionesis*, and *N. speciosa*. Ultimately, Haviland returned the plant to the *Mitragyna* genus; thus, it was accepted as *M. speciosa*.

## 4. Habitat and Cultivation of *Mitragyna* Plants

Naturally, *Mitragyna* species can grow in swampy habitats, riverbanks, inundated soils, or non-inundated soils [52]. Plants of this species are generally tree forms that do best in areas with wet or moist soil and also do well on rocky soils with moderate to total sun exposure [53]. *Mitragyna* plants thrive in the rainy season, and their leaves will fall off during the dry season. In less fertile to average soil conditions, these plants can grow to a height of 4–9 m. However, if the place where they grow is on fertile soil or in their natural habitat, they can grow to a height of 15–30 m [43,53,54,55]. *Mitragyna*’s habitat is in watersheds and swamps. *Mitragyna* plants (mainly *M. speciosa*) grow optimally in alluvial soil (mineral deposits) that is fertile with sufficient water. This plant has the ability to survive under waterlogged conditions.

The above conditions can be found in several areas in Indonesia, such as land areas in Kalimantan. *M. speciosa* is mostly found growing naturally in the same habitat as aquatic plants such as *Donax canniformis*, *Limnocharis flava*, *Ipomoea aquatica*, and *Nauclea officinalis*. *N. officinalis* has a morphology similar to *M. speciosa*, and it is often used as an adulterant of this plant. *M. speciosa* is easy to grow from seeds that fall from trees and breeds quickly in moist soil. In addition, this plant also functions as a barrier to soil erosion on the banks of rivers.

The genus *Mitragyna* is very easy to cultivate. *M. speciosa* is one of the most widely cultivated species in this genus. In several areas in Indonesia (especially in Kalimantan), local people cultivate and trade the leaf part of this plant as an export commodity, which has become the major source of income for farmers. This plant can easily be cultivated under various conditions, such as moist, dry, or swampy soils. However, apart from cultivation, these plants still grow naturally in forest areas, swamps, and on riverbanks [56]. Zhang et al. (2020) [57] reported that the fertility level of soil where *M. speciosa* grew had little effect on the contents of secondary metabolites (mainly alkaloid content) in the leaf. Meanwhile, low-to moderate fertilizer levels increase the levels of several compounds, such as isocorynantheidine, corynantheidine, and speciogynine. Fertilizer is vital to further research to determine the optimum fertilization conditions to increase the levels of target secondary metabolites. In addition, some local farmers (in East Kalimantan) claim that irrigation also affects the fertility level in the environment where this species is grown.

The leaves of *M. speciosa* are harvested for the first time when the plants are six months old with an average height of ±1 m. Harvesting is carried out by picking the old leaves and leaving the shoots and a few young leaves. The next harvest is carried out about 1–3 months after the first harvest. The second and subsequent harvests will continue to experience an increase in production of about 30%. The level of soil fertility strongly influences this increase. *M. speciosa* plants that are large and about 2.5 years old have the main stem cut out to facilitate harvesting and stimulate the emergence of branching, cutting the height to 1.5, 1, or 0.2 m from the base of the stem [58].

## 5. The Genus *Mitragyna* in Traditional Medicine Uses

*Mitragyna* is traditionally used in several countries where widely grown, such as in Africa and Asia. This genus has been used from generation to generation for hundreds of years, and some species were even recorded in traditional medicinal systems.

In Africa, most species of this genus are used by local people to treat various diseases; *M. ledermanii*, *M. stipulosa*, *M. inermis,* and *M. rubrostipulosa* are traditionally used to treat malaria [59], rheumatism, cardiovascular diseases, amenorrhea, inflammation, headaches, bronco-pulmonary diseases, and diabetes mellitus [55]. The stem bark of this plant is traditionally combined with *Garcinia cola* to treat trypanosomiasis in Nigeria [60]. *M. inermis* leaf decoction is used as an antidiabetic in Ivorian traditional medicine [61] and is used to treat inflammation [62].

In Asia, the genus *Mitragyna* is also widely used in traditional ways, as in Africa. Several species are widely used in traditional medicine, such as *M. parvifolia*, *M. hirusta*, *M. diversifolia*, *M. rotundifolia*, and, controversially, *M. speciosa*. As for the species of *M. tubulosa*, empirical use data were not recorded in the search for articles from 1990–2021, and research publications on this species are still very limited. *M. parvifolia* is empirically used as an anthelmintic [55] and antifungal [63]. In Ayurveda, traditional medicine practitioners use bark and roots to treat fever, colic, burning, poisoning, muscle aches, edema, coughs, gynecological disorders, and as aphrodisiacs. The leaves are used to treat wounds by attaching them to reduce pain and swelling. Fruit juice is used to increase breast milk, as a depurant [63,64], and as an anticonvulsant [65]. *M. rotundifolia* leaves [66] and *M. diversifolia* bark [20] were empirically used for antidiarrhea, mainly in the Bawm tribe in Bangladesh. The leaves of *M. hirusta* are used as an analgesic and to treat mental illness [55].

*M. speciosa* is a controversial endemic Asian plant of this genus, although it has long been used in traditional medicine. In Thailand, Malaysia, Indonesia, and several neighboring countries, the leaves help reduce headaches, pain, and toothaches and increase stamina. In addition, it was also noted that *M. speciosa* was traditionally used for hypertension, diabetes, and fever. Traditional methods of administration include chewing, smoking, or regular decoction [5,38,67]. The use of *M. speciosa* in males has been studied in rural communities in southern Thailand and has been shown to increase work productivity, make people stronger, less sleepy, less tired, more energetic, and more active. The number of leaves chewed or brewed also varies, with a single dose of 0.5–4 leaves, or 10–80 leaves per day for regular users. Some users who have routinely consumed this species for 25, 20, and seven years can stop using *M. speciosa* for varying lengths of time. Moreover, users consume it in the form of fresh leaves, and it is also used in the form of tea drinks made from dried leaves. Users can preserve the leaves in the dry form if the plant is abundant because the effects are the same in both the fresh and dried forms [68]. In addition, it is also reported to be used as an aphrodisiac [69].

In Thailand, *M. speciosa* is more commonly used by men than women. Respondents in Assanangkonchai’s (2007) [70] study believe that kratom can improve work performance by increasing resistance and tolerance to sunlight and by helping the user overcome fatigue. In particular, *M. speciosa* is also known to treat several conditions such as diarrhea, cough, and hypertension [55]. A similar ethnopharmacological study was also carried out in northern Malaysia by Ahmad and Azis (2012) [71]. The main reasons for using *M. speciosa* are stamina and endurance, social and recreational use, and improving sexual performance. In addition, based on the primary reasons for using *M. speciosa* provided by users/respondents, *M. speciosa* is used to facilitate sleep, for its euphoric effects, and to reduce symptoms of withdrawal from opioid drugs [71]. In Singh’s (2015) study regarding the socio-demographics of *M. speciosa* users, it is known that 83% of users know that the plant can cause addiction and dependence [67].

*M. speciosa* has traditionally been used in Indonesia to improve stamina, cure pain, rheumatism, gout, hypertension, stroke symptoms, diabetes, sleeplessness, wounds, diarrhea, cough, cholesterol, typhoid, and stimulate appetite [5,58,72,73]. Several empirical properties have been studied, including substantial analgesic effects, sedative effects, immune system boosting benefits, stimulant and anti-depressant effects, usage in pregnant and nursing women, and possible misuse and withdrawal symptoms [10,15,26,74,75,76]. The action are caused by secondary metabolite contains in *M. speciosa* leaves (mainly mitragynine and 7-hydroxymitragynine), which are later accountable for these pharmacological effects [77,78,79,80,81,82]. In addition, there are many other groups of secondary metabolites [21,50,80,83,84,85] that are interesting for further research.

## 6. Phytochemistry of the Genus *Mitragyna*

Research on the phytochemical content of the genus *Mitragyna* has been widely reported for hundreds of years. As with other genera from the Rubiaceae family, the genus *Mitragyna* (mainly *M. speciosa*) has been reported to be rich in compounds of the alkaloid group [3,23,78,79,80,81,82]. So far, as many as 79 secondary metabolites from the genus *Mitragyna* have been published and described in detail, including the dominant compounds of alkaloids, flavonoids, and polyphenols [21,80], terpenoids, triterpenoids [83], saponins, and secoiroids [50,84,85] as shown in Table 2.

Brown et al. (2017) [50] conducted a literature survey, noting as many as 57 secondary metabolite compounds, 37 of which were unique groups of alkaloids derived from *M. speciosa*. Likewise, Firmansyah et al. (2021) [24] explained that the leaves of *M. speciosa* contained more than 40 types of alkaloids, with six of them having been identified and tested to be pharmacologically active, especially as a psychoactive substances, including mitragynine, 7-hydroxymitragynine, mitraphylline, speciociliatine, speciogynine, and paynantheine (as shown in Figure 1). The most abundant compound found in commercial kratom products (the leaf of *M. speciosa*) is mitragynine, which contains 2% of the total weight and 66% of the total alkaloid content of the dried *M. speciosa* leaf, whereas 7-hydroxymitragynine is an active metabolite oxidized from mytraginine at 0.02% of the total weight of the dried *M. speciosa* leaf [24].

Previous phytochemical studies on the *Mitragyna* genus isolation found indole alkaloids (mainly mitragynine) [94,99] and triterpenoid saponins [23,103,117] to be the predominant ingredients of this plant, accounting for roughly half of the total alkaloid content. At the same time, in vitro and/or in vivo studies, as well as clinical trials, have shown that *Mitragyna* and its active components have broad pharmacological activities.

## 7. Pharmacology and Toxicity of the Genus *Mitragyna*

The *Mitragyna* genus has been used as a traditional medicine to treat diseases [99,100]. In this section, we discuss the pharmacological and toxicological aspects of *M. speciosa*. As it gained popularity in recent years, *M. speciosa* consumption also increased, raising concern for its possible development as a beneficial natural remedy and its potential risk of abuse among the public.

### 7.1. Pharmacological Aspect of M. speciosa

*M. speciosa* contains more than 40 compounds of alkaloids, four of which have been proven to be active, i.e., mitragynine, 7-hydroxy mitragynine, speciociliatine, and corynanthidine [23,24]. Mitragynine is the most abundant component in the *M. speciosa* leaf, constituting 66% of the total alkaloid content, whereas others only account for 1–9% [53,118]. Mitragynine is an indole-containing alkaloid and is suggested to have approximately 13 times the potency of morphine [53,118,119].

Idayu et al. (2011) [13] reported that the administration of mitragynine at doses of 10 mg/kg and 30 mg/kg (intra peritoneal) was able to provide an antidepressant effect on mice, as suggested by the results of two behavioral experiments, the mouse force swim test and the tail suspension test. Overall, the results of this study are consistent with the results obtained in a clinical setting, further implying that mitragynine plays a role in the regulation of depression and may have psychotherapeutic value in the treatment of depressive disorders [13]. However, it is important to note that both the mouse force swim test and the tail suspension test are not specifically established to model the depression condition and there are some differences between experimental animal and clinical studies in humans. Hence, the results obtained from these experiments need to be carefully viewed and interpreted.

Reanmongkol et al. (2007) [10] demonstrated the antinociceptive response of mice given methanol and alkaloid extracts in hot plate tests; the mice were suspected of having activity in the supraspinal system. It has been reported that mitragynine, the active alkaloid in *M. speciosa* leaves, has antinociceptive actions on noxious mechanical stimulation by involving the descending noradrenergic and serotonergic systems of the supraspinal opioid system [76,120], which is primarily mediated by mu- and delta-opioid receptor subtypes in mice [121] and has a similar effect to oxycodone and morphine [11]. In addition to its opioid-like analgesic effect, mitragynine stimulates postsynaptic alpha-2 adrenoreceptors and inhibits cyclooxygenase-2 messenger RNA (mRNA) and protein expression, suggesting non-opioid receptor pain-relieving effects [53,118,122]. The inhibition of COX-2 may affect the formation of PGE2, which can lead to anti-inflammatory effects [6,53,123]. Moreover, mitragynine impairs neuronal pain transmission via the blockade of Ca^2+^ channels, which has been proposed as another antinociceptive mechanism of *M. speciosa* [124].

Methanolic extract from the *M. speciosa* leaf has an antidiarrheal effect by reducing diarrheal frequency, the total diarrheal score, fecal weight, and intestinal transit. The proposed mechanisms were through anti-permeability action and decreased gastrointestinal motility. The enteric nervous system controls motility in the small intestine primarily through excitatory and inhibitory impulses. However, these local nervous system signals are modulated by inputs from the central nervous system, and many gastrointestinal hormones appear to affect intestinal motility. These links are parasympathetic and sympathetic fibers that connect the central nervous system directly with the digestive tract. Thus, the extract of *M. speciosa* leaves may exert its effect on other pathways in addition to the effect of mitragynine on the opioid receptors [19].

As the primary alkaloid in *M. speciosa*, mitragynine undergoes metabolism in humans via phase I and II mechanisms. After the parent drug is hydrolyzed, it undergoes O-demethylation followed by oxidative and reductive transformation to form intermediate aldehydes and conjugate glucuronide formation as the final step of phase II metabolism [53,118]. The half-life of mitragynine is about 3.5 h and it is eliminated from the body mainly in the urine [118,125].

The onset of the effects of *M. speciosa* is about 10–20 min, and the full effects can be achieved about 30–60 min after ingestion. The strongest effects of *M. speciosa* occur at about 2–4 h. However, generally, the effects can last approximately 5–7 h after consumption and will weaken after 24 h [118,119,126,127]. The extract of *M. speciosa* inhibits various cytochrome P450 (CYP) enzymes, notably, CYP 3A4, 2D6, and 1A2. Prescribed or over-the-counter (OTC) medicine should be taken with caution because clinically significant interactions may appear [128].

Even though *M. speciosa* has potential beneficial health effects, these are based on case reports, preclinical animal studies, and derived from its traditional use in Southeast Asia. The effect of the real use in humans to self-treat acute or chronic pain and other psychiatric conditions was obtained by a user-based survey [129,130,131]. This method has limitations in outcomes and health diagnosis since it is self-reported and includes small sample sizes for specific *M. speciosa* uses and health conditions.

### 7.2. Toxicological Aspect of M. speciosa

*M. speciosa* can be used against fatigue and can produce stimulant effects in low to high dosages [33,132]. Consumption of this plant under long-term and high-dose conditions could lead to several atypical effects and other effects such as anxiety, irritability, and enhanced aggression [69]. Frequent users report tremors, anorexia, weight loss, seizures, and psychosis [53,118]. In addition, seizures and addiction are mainly experienced by individuals following long-term *M. speciosa* consumption, and liver toxicity can occur after *M. speciosa* overdose [118,133]. Individuals with long-term addiction to *M. speciosa* have been reported to have hyperpigmentation of the cheeks, tremors, anorexia, weight loss, and psychosis [33].

Several case report studies have reported *M. speciosa* consumption as being linked to medical conditions and death [134], including a case report of a 32-year-old male who was found having seizure-like movements and foaming at the mouth. Despite the administration of benzodiazepine and intubation, the patient’s movement persisted. Twenty-four hours after extubation since the first treatment, the patient admitted to the consumption of *M. speciosa*, obtained from the internet [135]. Another seizure effect after *M. speciosa* ingestion was reported in a case report of a 64-year-old male, followed by a period of unresponsiveness. The detected mitragynine concentration in the urine was 167 ± 15 mg/mL [136].

A 44-year-old subject bought *M. speciosa* leaves from the internet and consumed them to overcome chronic abdominal pain. The patient developed a myxedematous face and lethargy following opiate withdrawal syndrome. Severe primary hypothyroidism occurred after consuming *M. speciosa* leaves for four months. However, this condition improved after fifteen months of taking oral opiates (methadone and oxycodone) combined with levothyroxine [137].

One case reported a 25-year-old man with jaundice and pruritus who was admitted to the hospital after ingesting *M. speciosa* excessively for two weeks. Drug-induced intrahepatic cholestasis was identified after liver biopsy. Mitragynine was confirmed after both urine and serum samples were examined [133].

In animal models, particularly in rats, Azizi et al. (2010) [138] reported lethal effects of 200 mg/kg total alkaloid extract of *M. speciosa*. In contrast, Janchawee et al. (2007) [139] reported the same effect after an oral dose of 200 mg mitragynine. Spontaneous behavior, food and water consumption, absolute and relative organ weight, and hematological parameters did not change after acute oral administration of 100, 500, and 1000 mg/kg doses of a standardized methanolic extract of *M. speciosa*. However, the administration of methanolic extract led to a substantial rise in alanine transaminase (ALT) and argininosuccinate lyase (ASL), and elevated creatinine as a sign of nephrotoxicity was also seen at the highest (1000 mg/kg) dose [140]. The impairment of kidneys and lungs, over-inflation of the alveoli, and increased blood urea and serum creatinine were detected in sub-chronic high doses of *M. speciosa* [141].

In mice, the median lethal dose (LD_50_) for the oral administration of the methanolic and alkaloid extracts of *M. speciosa* was 4.90 g/kg and 173.20 mg/kg, respectively [10], which means that it has neither cytotoxic nor highly toxic effects. Chronic administration of mitragynine can lead to addiction and cognitive function impairment in mice [18,33].

## 8. The Abuse and Addiction of the *Mitragyna* Species

*M. speciosa* is a plant from the genus *Mitragyna* that has been widely reported for its abuse and addiction properties, upon consumption at high doses used (10–30 leaves daily). As has been previously reported, consumption of the fresh leaf of *M. speciosa*, about 3–10 times a day depending on the sensation of their need, was intended to overcome fatigue. In addition to fresh leaves, users or addicts may also consume the dried leaves after grinding [33]. Consumption of the dried leaf is usually either by a drink of warm liquid such as warm water or hot coffee or in the form of a self-made cigarette prepared from the dried leaves. Withdrawal symptoms that occur are similar to those of narcotic use. This may be related to the dose and duration of use [33]. The use of *M. speciosa* leaves was able to reduce pain and treat drug addiction. Several cases and studies in the United States and Southeast Asia can be used to reference this traditional medicine.

### 8.1. Abuse and Adverse Effects of M. speciosa

In the United States, Tabayali et al. (2018) [39] described a kratom user who developed a brief prodromal acute sickness, accompanied by increased liver enzymes with cholestatic characteristics and jaundice. In this aspect, kratom use is a crucial contributor to the spread of the opioid crisis, as therapeutic benefits are gained at the expense of potentially fatal adverse effects [39]. He had a liver disorder characterized by yellow skin accompanied by nausea, fatigue, joint pain, long night sweats, and high bilirubin values with detectable mitragynine concentrations in urine. The pharmacological effects of using *M. speciosa* leaves and their constituents depend on the usage dose. Low doses (1–5 g) may provide a mild stimulant effect to help the user with fatigue. Opioid-like effects such as analgesia, constipation, euphoria, and sedation are produced at moderate to high doses (5–15 g) [33,34].

In another case, a 64-year-old man had recurrent seizures after consuming *M. speciosa* leaf tea mixed with *Datura stramonium* for one month due to chronic pain after post-colostomy surgery [136]. Men also experienced tonic-clonic seizures 43 years after taking *M. speciosa* leaves and 100 mg of modafinil. The combination of *M. speciosa* with tramadol and propylhexedrine has been reported to cause death [142]. There were 11 deaths in the United States related to *M. speciosa* exposure, and all of them were among adults aged 22–38 years. Two deaths involved single-substance *M. speciosa* use, and nine other deaths involved exposure to multiple substances with agitation and tachycardia symptoms. *M. speciosa* abuse has been found to be on the rise in Texas from January 2009 to the present. Some cases are of *M. speciosa* alone and involve additives from the opioid and alcohol groups. The reported effects are clinical signs of potentially toxic abuse [35,36].

### 8.2. Addiction Effects of M. speciosa

The consumption of *M. speciosa* leaves in Thailand and Malaysia has been shown to improve quality of life, especially as part of postoperative treatment. For treating pain after surgery, *M. speciosa* is an excellent substitute for narcotic analgesics. However, the use of *M. speciosa* at high doses (in the form of juice or tea with doses exceeding more than three glasses of the dried leaf of *M. speciosa* daily) and for a long time (its use as nettle leaf supplements for more than five years) has the risk of causing addiction as in narcotics [143,144]. In observations using the Clinical Opiate Withdrawal Score (COWS), patients experienced a low-addiction reaction after withdrawal from *M. Speciosa* [145].

The average consumption of a high dose of *M. speciosa* leaves, such as three glasses per day of *M. speciosa* leaf tea with mitragynine content per cup, is 79 mg/glass, which shows an average intake of 276.5 mg daily. At the same time, the use of 350 mL of *M. speciosa* leaf juice containing 83.4 mg of mitragynine is reported to be a risk for depression due to withdrawal symptoms [146]. The risk of addiction is still lower compared to opioids, methamphetamine, marijuana, benzodiazepines, and ketamine [147]. As confirmed by the results of a study reported by Wilson et al. (2021) [148], administration of a combined alkaloid extract of *M. speciosa* or mitragynine showed minimal symptoms and produced a weaker dependence effect than the full agonist morphine.

### 8.3. Addiction Withdrawal Symptoms of Opioid Dependence

The alkaloid compounds of mitragynine and 7-hydroxymitragynine from the *M. speciosa* leaf are known to stop opioid addiction based on studies reported by Matsumoto et al. (2005) [124], Hassan et al. (2020) [149], and Harun et al. (2019) [150], using experimental animals. Matsumoto et al. (2005) [124] reported that the administration of 7-hydroxymitragynine (10 mg/kg, subcutaneous injection) twice daily for five days promoted lower withdrawal signs than the morphine-dependent group of mice (10 mg/kg, subcutaneous injection). Hassan et al. (2020) [149] found that mitragynine (5–30 mg/kg; intra peritoneal) can mitigate acute morphine dependence in rats. The withdrawal symptoms were significantly reduced after four days of mitragynine replacement, suggesting that mitragynine, similar to methadone and burprenorphine, can reduce morphine withdrawal symptoms. Similar results were reported by Harun et al. (2019) [150].

The administration of mitragynine (twice daily for 14 consecutive days at doses of 10 and 30 mg/kg) in rats could reduce the effect of morphine dependence in a better rate than buprenorphine at doses of 0.3 and 1.0 mg/kg. Mitragynine does not cause physiological dependence, but it does alleviate the physical symptoms of morphine withdrawal, which are desirable characteristics of novel pharmacotherapeutic interventions for treating opioid use disorder (OUD) [145]. Several years earlier, Vicknasingam et al. (2010) [69] reported similar results on the effect of symptom relief in a clinical observation study using a cross-sectional survey of 136 active users in the northern states of Kedah and Penang in Malaysia. The informal use of three glasses of *M. speciosa* (approximately 250 mL per glass) a day with a daily content of 67.5–75 mg mitragynine is significant for relieving symptoms of heroin dependence [69].

The potency of mitragynine and 7-hydroxymitragynine in *M. speciosa* is thought to be due to the agonist activity against mu- and kappa-opioid receptors, which can reduce opioid withdrawal symptoms [69,151]. Mitragynine and 7-hydroxymitragynine exhibit opioid-mediated (naloxone sensitive) antinociceptive activity. Both components share a similar structure to morphine, with three main functional groups binding to opioid receptors: tertiary nitrogen, benzene, and hydroxyl phenolic. The group can bind to both mu-receptors and kappa-receptors with an affinity for 7-hydroxymitragynine, 40 times stronger than mitragynine and 10 times stronger than morphine [25]. This may cause the dosage of *M. speciosa* to be lower than morphine to treat pain, so that withdrawal symptoms are lessened. The use of *M. speciosa* extract or single mitragynine showed withdrawal symptoms in mice that were lower in physical behavior than morphine on mu-receptor agonists. Likewise, this substance has been used to improve symptoms of morphine dependence [143]. The mytraginine content is lower in the Malaysian plants (12%) than in the Thailand ones (66%), which can also be attributed to the large number of people using the plant in Peninsular Malaysia, which has succeeded in reducing addiction due to long-term narcotic use [69].

Overall, the traditional use of *M. speciosa* as well as mtragynine alkaloid derivatives in laboratory experiments using experimental animals has shown a lower addictive effect on withdrawal symptoms compared to narcotics and can be used as therapy to reduce addiction due to the use of these narcotics. So far, the addictive effects that have been reported only occur in long-term users with high doses [143,144,145,146], while withdrawal symptoms occur in short-term users also at high doses [69,124,149,150]. Reports of addiction to the use of *M. speciosa* leaf can be a reference for the FDA of each country. Supervision and regulatory arrangements can be a solution for the potential utilization of *M. speciosa* commodities.

## 9. Legal Issues of *Mitragyna* Plants

One member of the species of the *Mitragyna* genus, *M. speciosa*, is of concern in various countries, especially regarding legal issues. *M. speciosa* is a widely available and unlisted herbal supplement that has been used in the treatment of opioid addictions, traditional medicine, and pain relief. *M. speciosa* preparations are available in pill, resin, herb, leaf extract, and leaf powder forms [5,50]. In 2008, the European Monitoring Centre for Drugs and Drug Addiction (EMCDDA) surveyed 27 European online sites selling “legal highs” and found that *M. speciosa* was one of the most widely offered and available drugs on 44% of the online sales sites investigated [152,153].

The wide availability of *M. speciosa* on the internet shows a wide public demand [153]. Since 2013, some dosage forms of the *M. speciosa* leaf, such as bottles of liquid preparations, were available for purchase in US head shops; however, this unique formula was no longer available in 2014. This change in availability is due to distributors and businesses implementing procedures for legal protection in response to changes in the law governing certain chemical components found in *M. speciosa*. The powdered form of the dried leaf of *M. speciosa* is currently the most readily available. When the powder this plant is added into teas and beverages, they have a gritty taste. Small packs of 60 g (1.5 oz) beverages (known as “Kratom Premix”) are available for purchase. Powder and liquid forms became available in Dutch “smart shops” in 2014 [153].

Until recently, the legal status of *M. speciosa* plants, psychoactive alkaloids, and their varying constituents worldwide was legal, illegal, and under government control [154]. In 2014, *M. speciosa* was legalized in several countries, including Austria, Belgium, Hungary, the Netherlands, the United Kingdom, and 43 states in the United States. Additionally, countries that prohibit *M. speciosa* use and its derivative compounds are Indonesia, Malaysia, Myanmar, Romania, Russia, and South Korea. Currently, dried *M. speciosa* leaves, mitragynine components, or other alkaloids from the plants are listed under the United Nations Drug Conventions of 1961 or 1971; hence, each country, state, or territory determines its legal position individually. The use of the mitragynine compound from *M. speciosa*, 7-hydroxymitragynine, is controlled in several countries, including Denmark, Finland, Germany, Latvia, Lithuania, Poland, New Zealand, Sweden, Vietnam, Romania and several states of the United States [29,30,31,73,154,155,156].

Governments have implemented various methods to try to control *M. speciosa* or its alkaloids within their legislative framework. For example, in Thailand, *M. speciosa* has been illegal since 1943 and is held under the 1979 Narcotics Act alongside cannabis and other psychotropic mushroom species. However, this plant was taken off the list as a narcotic on 24 August 2021.

In Malaysia, mitragynine, the primary alkaloid of the *M. speciosa* leaf, is regulated under the Poisons Act 1952, and individuals who violate this law can be fined a maximum of RM 10,000 and can receive up to four years of imprisonment. In the United States, *M. speciosa* is not scheduled under the Controlled Substances Act. However, drug information on *M. speciosa* is available on the DEA website, which states there are no legal medical uses for *M. speciosa* in the United States. Therefore, *M. speciosa* leaf and its derivatives (mitragynine and 7-hydroxymitragynine) in several different states, among others, are prohibited from being legally advertised as a treatment, their ownership is prohibited, and their sales are limited to particular population groups [154].

Australian law strictly prohibits the unauthorized sale, distribution, use, and manufacture of *M. speciosa* and mitragynine. According to Schedule 9 (S9) of the Standards for the Uniform and Scheduling of Medicine and Poisons (SUSMP), *M. speciosa* may only be used for research purposes. Under the 2009 Medicines Amendment Regulations in New Zealand law, the sale of *M. speciosa* leaves to individuals without a doctor’s prescription is illegal. Therefore, certain people interpret importing and owning *M. speciosa* leaf products as technically illegal in New Zealand [154].

Further information on the legal status of *M. speciosa*, mitragynine, and their derivatives starting from 2014 can be seen in Table 3. The legal status of countries not listed in Table 3 cannot be verified through primary sources [153,154].

Globally, Table 3 shows that the regulatory system related to the *M. speciosa* leaf is divided into three groups, namely, (1) countries that do not prohibit the circulation and use of *M. speciosa* leaf commodities; (2) countries that do not prohibit the circulation and only restrict the use of this commodity; and (3) countries that prohibit its circulation and use. According to Wahyono et al. (2019) [58], the difference in regulating the circulation and use of this commodity is due to the lack of strong scientific evidence regarding generalizable positive and negative impacts of its use in the health sector [58].

In Indonesia, *M. speciosa* leaves are prohibited as a raw material for traditional medicines and food supplement products. However, no regulation prohibits the cultivation, distribution, and/or marketing in the form of fresh or dried leaves or extracts. *M. speciosa* is also a potential crop in agriculture based on the regulation of the Minister of Agriculture No. 104 of 2020. On the other hand, the National Narcotics Board (BNN) of the Republic of Indonesia will continue to target the rules prohibiting the distribution and use of *M. speciosa* leaves starting in 2022, even though the regulation does not yet exist. The information from BNN has been widely discussed until now, becoming controversial and a source of anxiety among farmers and exporters.

## 10. The Prospective Potential of *Mitragyna* Species as a Pharmaceutical Product

In general, plants of the genus *Mitragyna* (especially *M. speciosa*) are well known and have high economic value. The leaf part of this plant is a widely available herbal supplement that is sold in various forms of dried leaves (powder or chopped), pure or concentrated extracts, and liquid preparations (energy drinks) in Western countries, and the forms are generally taken orally [168]. The trade-in *M. speciosa* leaf powder has existed in Indonesia since the 2000s, but it has only flourished in the last five years, and it has even become a prime commodity for export by farmers in several areas on the Indonesian island of Kalimantan.

Schmidt et al. (2011) [169] investigated product prices and the availability of this plant on 314 online sales sites under “legal highs” and reported price ranges of €6–15 (=£4.70–11.70, $7.60–19) per 10 g (=0.35 ounce) of dry *M. speciosa* leaf and €7.50 (=£5.50, $9) per gram of “kratom 15X” in the extract. According to a report from the DEA, dried *M. speciosa* leaf retails for €8–31 (=£6–24, $10–40) per ounce (=38 g) in the United States. The leaf powder form (commercial grade white Vein Thai, Green Malaysia, and Bali) sells for €4.70 (=£3.70) and $6 per ounce (=38 g) on websites that marketed online in 2014. According to the Maeng Da website (https://web.asu.edu/educationblog/maeng-da-kratom-origin-%E2%80%93-types-and-benefits, accessed 12 October 2021), high-quality products are available for €6–14 (=£511, $818) per ounce (=38 g). Based on the latest data quoted from website: www.misterexport.com (accessed 12 October 2021), the selling price of domestic *M. speciosa* leaf powder is Rp. 98,000 or the equivalent of about USD 6–7 per kg. Meanwhile, the selling price abroad varies between USD 17–25 per kg, depending on the quality, with several export destination countries such as South Korea, America, Canada, Singapore, Japan, India, UAE, China, Hongkong, Saudi Arabia, and Kuwait [58,170].

*M. speciosa* has long been used to improve the stamina of outdoor laborers (farmers) and for mood enhancement, pain relief, and opium addiction. Surprisingly, this plant has been found to have a paradoxical effect in that it can provide both stimulant and sedative effects depending on the dosage. The *M. speciosa* leaf contains many biologically active alkaloids [171]. Because of the popularity of this plant, several studies have focused on its use, especially on the interaction of mitragynine, which has an opioid-like effect. Given the large number of cases of its abuse and the side effects caused by its use that have been reported in Europe and America, this is in stark contrast to the cases reported under its traditional use in Southeast Asia for centuries [172,173,174].

The overall physiological action of *M. speciosa* as a pharmacological agent is complicated since it involves intermixing stimulant and opiate-like qualities in a dose-dependent manner due to the diversity of alkaloids contained in kratom extracts and the distinct potential pharmacologic capabilities of each (primarily stimulant-like at low doses, with opioid effects predominating at higher doses) [5,151]. In addition, other compounds such as flavonoids and polyphenols [108] in methanol extract have antioxidant and antimicrobial activity, as reported by Parthasarthy et al. (2009) [21]. The same extract also has anti-inflammatory and antinociceptive activity [6].

However, this plant has potential prospects for raw materials as a promising pharmaceutical product despite all the controversy. In line with the development of extraction technology with a high degree of selectivity, modern approaches are essential in developing more effective, selective, and high-yielding methods for target secondary metabolites [175]. In addition, a non-conventional green solvent-based extraction method has been developed, namely, the natural deep eutectic solvent-based microwave-assisted extraction method [176,177] and ultrasound-assisted extraction [178]. This method is very effective in attracting the target secondary metabolite compounds, mainly polyphenols and other non-alkaloid compounds. The successful development of this modern extraction method has become one of the main keys to developing pharmaceutical products [176,177,178].

On the other hand, the development of pharmaceutical products with active ingredients derived from natural products (mainly from plants) has also increased. The use of herbal medicine is now in great demand, especially in developing countries, for primary health care, not only because it is cheap, but also because of its high level of cultural acceptance, compatibility with the human body, and minimal side effects. However, the use of the *M. speciosa* leaf as a source of raw materials for herbal medicines has recently been highly controversial considering the many reports related to the effects caused by its use. Therefore, to be accepted as a viable alternative to modern medicine, it is necessary to develop valid pharmaceutical products that have been proven to be safe and effective. The utilization of the natural deep eutectic solvent-based microwave-assisted extraction method and non-conventional ultrasound-assisted extraction, or known as “a green extraction method approach,” with high selectivity in extracting the target compound of secondary metabolites is hoped to respond to the controversy. Given the potential pharmacological activity described above, *M. speciosa* strongly correlates with empirical/traditional use in society. With the application of selected extraction methods, the resulting product can increase the effectiveness of pharmaceutical products with active *M. speciosa* leaves by minimizing unwanted effects.

## 11. Conclusions

In conclusion, plants of the *Mitragyna* genus (mainly *M. speciosa*) are easy to cultivate, traditionally have many benefits for treating various diseases, and have high economic value. However, these plants contain several biologically active alkaloid compounds, such as opioids, which can provide a stimulant effect and cause dependence, so there is a chance for abuse. On the other hand, *M. speciosa* has future potential as a pharmaceutical product, with dosage adjustment and use for oral consumption with various pharmacologically proven properties. Therefore, taking into account the potential benefits and economic value of this species, it is highly recommended for the preparation of government policies in terms of regulating and supporting studies on the development of separation techniques for compounds that have the potential for abuse and addiction so that its potential biological benefits as drugs, as well as its potential for increasing economic income from cultivation by local communities, can be optimized.

## Figures and Tables

**Figure 1 life-12-00193-f001:**
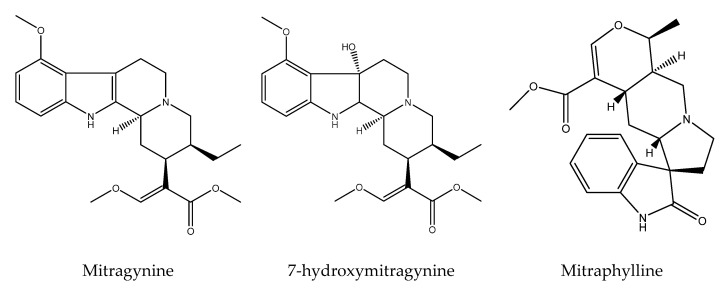
*M. speciosa* leaves contain physicochemically active indole and oxindole alkaloid compounds.

**Table 1 life-12-00193-t001:** The botanical origin of the plants of the genus *Mitragyna*.

No.	Species	Synonyms	Botanical Origin	References
1	*M. ledermannii* (K. Krause) Risdale	*M. ciliate* Aubrev. & Pellegr.: *Fleroya ledermannii* (K. Krause) Risdale Y.F.Deng.; and *Hallea ciliate* (Aubrev. & Pellegr.) J.F.Leroy	The species spreads from eastern Liberia to the Central African Republic, the south of Gabon, Congo, and Angola	[48,49,50]
2	*M. rubrostipulata* (K. Schum) Havil.	*F. rubrostipulata* (K. Schum) Y. F. Dengs; *H. rubrostipulata* (K. Schum.) J.F.Leroy; and *Adina rubrostipulata* K. Schum.	The species spreads across various regions in Africa, including the Democratic Republic of Congo, Ethiopia, Tanzania, Malawi, and Mozambique	[44,47,49,50]
3	*M. inermis* (Willd.) Kuntze	*M. africanum* (Willd.) Hook; *Nauclea africana* Willd.; *N. africana* var. *luzoniensis* DC.; *N. platanocarpus* Hook.f.; *N. inermis* (Willd.) Baill; *Cephalanthus africanus* Rchb.; *Platanocarpum africanum* (Willd.) Korth; *Uncaria inermis* Willd.; and *Adina inermis* (Willd.) Roberty	The species spreads across eastern Mauritania to Sudan	[44,47,49,50]
4	*M. stipulosa* (DC.) Kuntze	*M. chevalieri* K. Krause.; *M. macrophylla* Hiern.; *F. stipulosa* (DC.) Y.F. Deng.; *N. stipulosa* DC.; *N. bracteosa* Welw.; *Adina stipulosa* (DC.) Roberty; *Mamboga stipulosa* (DC.); and *H. stipulosa* (DC.) J.F. Leroy	The species spreads from eastern Senegal to Uganda and southern Senegal to Zambia and Angola	[44,49,50]
5	*M. hirusta* Havil.	*M. Africana* (Willd.) Korth.; *Platanocarpum Africana* (Willd.) Hook.; *Cephalanthus africanus* Rchb.; *N. africana* Willd.; and *Paradina hirusta* (Havil.) Pit	The species is found in the Asian region, mainly in Thailand, Vietnam, Laos, China, and Cambodia	[43,44,49,50]
6	*M. diversifolia* (Wall. Ex G.Don) Havil.	*M. javanica* Koord; *Stephegyne parvifolia* Vidal; *S. tubulosa* Fern.; *N. diversifolia* Wall. Ex. G. Don; *N. adina* Blanco; and *Mamboga capitata* Blanco	This species spreads across Asia, i.e., Indonesia, Malaysia, the Philippines, Thailand, Vietnam, Cambodia, Laos, and China	[43,44,49,50]
7	*M. parvifolia* (Roxb.) Korth.	*Stephegyne parvifolia* (Roxb.) Kuntze; *N. parvifolia* Roxb.; and *N. parvifolia* Willd	The species has been found in Asia, especially in Myanmar, Sri Lanka, India, and Bangladesh	[43,49,50]
8	*M. rotundifolia* (Roxb.) Kuntze	*M. brunonis* (Wall. Ex. G. Don) Craib.; *N. rotundifolia* Roxb.; *N. brunonis* Wall, Ex. G. Don.; and *Bancalus rotundifolius* (Roxb.) Kuntze	This species has been found in Asia, mainly in the regions of Thailand, Myanmar, Laos, China, India, and Bangladesh	[43,44,49,50]
9	*M. tubulosa* (Arn.) Kuntze	*N. tubulosa* Arn	The species is endemic to Asia (mainly India) and has spread to Kerala, Tamil, Nadu, and Sri Lanka	[43,49,50,51]
10	*M. speciosa* (Korth.) Havil	*N. luzoniensis* Blanco, *N. korthalsii* Steud, *N. speciosa* (Korth.), and *Stephegyne speciosa* Korth	The species is endemic to southeastern Asia and is scattered across various regions of Myanmar, Vietnam, Thailand, Malaysia, Indonesia, and Papua New Guinea	[40,49,50,52]

**Table 2 life-12-00193-t002:** Secondary metabolite compounds in leaves of the genus *Mitragyna*.

No	Species	Indole and Oxindole Alkaloids	Other Compounds	References
1	*M. ledermannii* (K. Krause) Risdale	Mitraciliatine, rhynchophylline, rhynchociline, ciliaphylline, rotundifoline, isorotundifoline	-	[2,50]
2	*M. rubrostipulata* (K. Schum) Havil.	Hirsuteine, mitraphylline, isomitraphylline, isorotundifoline, rotundifoline N- oxide, isorhynchophylline, rhynchophylline N-oxide, rhynchophylline, rotundifoline,	-	[50,86]
3	*M. inermis* (Willd.) Kuntze	Uncarine D (speciophylline), rhynchophylline, isorhynchophylline, rotundifoline, isorotundifoline	Quercetin, dihydrodehydrodiconiferyl alcohol, isolariciresinol, isolariciresinol-3α-O-β-D-glucopyranoside, ursolic acid, oleanoic acid, betulinic acid, barbinervic acid, quinovic acid and its derivates, inermiside I, inermiside II,	[2,50,87,88,89,90]
4	*M. stipulosa* (DC.) Kuntze	Mitraphylline, rhyncophylline, isorhynchophylline, rotundifoline, isorotundifoline	Ursolic acid, quinovic acid and its derivates, sitoseterol, stigmasterol, daucosterol	[2,50,84,91]
5	*M. hirusta* Havil.	Mitraciliatine, mitraphylline, isomitraphylline, isomitraphylline N-oxide, rhynchophylline, isorhynchophylline, isopteropodine, isomitraphyllinol, hirsuteine, mitrajavine, uncarine D (speciophylline), rhynchophylline, isorhynchophylline, rotundifoline, isorotundifoline	-	[50,92,93,94,95]
6	*M. diversifolia* (Wall. Ex G.Don) Havil.	7- hydroxy-isopaynantheine, 3-dehydro-paynantheine, 3-isopaynantheine-N(4)- oxide, mitrafoline, mitradiversifoline, specionoxeine-N(4)-oxide, specionoxeine-N(4)-oxide	3α, 6β, 19α-trihydroxy-urs-12-en-28-oic acid, 3β, 6β, 19α- trihydroxy-urs-12-en-28-oic acid; 3-oxo-6β-19α-dihydroxy-urs-12-en-28-oic acid; 3β, 6β, 19α-trihydroxy-urs-12-en-24, 28-dioic acid 24-methyl ester; 3β, 6β, 19α, 24-tetrahydroxy-urs-12-en-28-oic acid; rotundic acid; 23-nor-24-exomethylene- 3β, 6β, 19α-trihydroxy-urs-12-en-28-oic acid; pololic acid	[50,96,97,98]
7	*M. parvifolia* (Roxb.) Korth.	Dihydrocorynantheol, dihydrocorynantheol N-oxide, akuammigine, akuammigine N-oxide, 3-isoajmalicine, mitraphylline, isomitraphylline, rhynchophylline, isorhynchophylline, rotundifoline, isorotundifoline, speciophylline N-oxide, uncarine F, uncarine F N-oxide, pteropodine, isopteropodine, uncarine D (speciophylline), 16,17-dihydro-17β-hydroxy isomitraphylline, 16,17-dihydro- 17β-hydroxy mitraphylline	-	[50,64,99,100]
8	*M. rotundifolia* (Roxb.) Kuntze	mitraphylline, isomitraphylline, rhynchophylline, isorhynchophylline, isorhynchophylline N-oxide, rotundifoline	3,4-dihydroxybenzoic acid, cathecin, caffeic acid, epicathecin, kaempferol, 4′-O-methyl-gallocatechin, 4-hydroxy-3-methoxybenzoic acid, 3-hydroxy-4-methyloxybenzoic acid, cincholic acid, quinovic acid and its derivates	[50,99,101,102,103,104]
9	*M. tubulosa* (Arn.) Kuntze	Mitraciliatine, rhynchociline, ciliaphylline, rotundifoline, isorotundifoline, rhynchophylline, isorhyncophylline, mitraphylline, isomitraphylline, ciliaphylline N-oxide	-	[2,50,91]
10	*M. speciosa* (Korth.) Havil	mitragynine, 7-hydroxymitragynine, paynantheine, mitralactonal, mitragynaline, speciociliatine, speciogynine, mitrasulgynine, 3,4,5,6-tetradehydromitragynine, mitragynaline, mitragynalinic acid, corynantheidinaline, corynantheidinalinc acid, 3-dehydromitragynine, 9-methoxymitralactonine, 3-isopaynantheine, ajmalicine, isocorynantheidine, mitragynine pseudoindoxyl, mitraphylline, mitragynine oxindole A, mitragynine oxindole B, corynoxine, corynoxine B, mitraciliatine, 7β-hydroxy-7H-mitraciliatine, isomitraphylline, rhynchophylline, rhyncocilline, cilaphylline, isospeciofoleine, isospeciofoline, isorotundifoline	Apigenin, apigenin 7-glycosides, quercetin, quecitrin, rutin isoquercitrin, hyperoside, quercetin-3-galactoside-7-rhamnoside, kaempferol, kaempferol 3-glucoside, epicatecin, caffeic acid, chlorogenic acid, 1-O-feruloyl-β-D-glucopyranoside, benzyl-β-D-glucopyranoside, quinovic acid and its derivates, monoterpenes 3-oxo-α-ionyl-*O*-β-Dglucopyranoside, roseoside, secoiridoid, vogeloside, epigeloside	[3,23,50,71,72,105,106,107,108,109,110,111,112,113,114,115,116]

**Table 3 life-12-00193-t003:** The legal status of *M. speciosa* in some countries in the world.

Country	Status	Details
Austria [157]	Legal	
Belgium [157]	Legal	
Hungary [157]	Legal	Not approved for human consumption, but available as incense in head shops
The Netherlands [157]	Legal	Available in head shops
United Kingdom [157]	Legal	Sold in head shops (smart shops)
Thailand [157,158,159]	Legal	Thailand is considering making *M. speciosa* legal again to find safer and healthier stimulants to combat Thailand’s high rate of methamphetamine addictions. This plant was formerly listed as a narcotic in Thailand; the change took effect on 24 August 2021.
United States	Varying regulation legal or regulated in most states	Banned in the state of Indiana [160]. The state of Louisiana [161] prohibits the distribution of products containing *M. speciosa* to minors (under age 18). It is controlled and illegal to sell in the state of Tennessee [162]. In 2005, US Drug Enforcement Agency (DEA) listed this as a drug of concern with abuse potential [163,164] starting in 2014. *M. speciosa* is not a scheduled or restricted drug at the federal level
Russia [146,165]	Illegal	Mitragynine (9-methoxy-corynanthidine) and its derivatives are illegal
Malaysia [156,166]	Illegal	Controlled under narcotic law
Myanmar [157]	Illegal	Controlled under narcotic law
South Korea [156]	Illegal	
Indonesia [167]	Illegal	*M. speciosa* plants and their processed products, including active chemical compounds, are included in Narcotics Group I, and are stipulated under the Regulation of the Minister of Health, with a maximum transition period of five years, since 2020. *M. speciosa*, containing the alkaloid mitragynine at high doses, can have a sedative effect. It is classified as a narcotic and is included in the list of ingredients that are prohibited for use in dietary supplements and traditional medicines.
Burma [156]	Controlled	
Denmark [157]	Controlled	
Finland [157]	Controlled	Requires a prescription. Shipments can be seized at the border
Germany [157]	Controlled	Controlled as an approved pharmaceutical drug.
Latvia [157]	Controlled	
Lithuania [157]	Controlled	
Poland [157]	Controlled	
Sweden [157]	Controlled	
Vietnam [156]	Controlled	
Romania [156,157]	Controlled, illegal	
New Zealand [156,157]	Controlled, restricted	*M. speciosa* and mitragynine are controlled under Schedule 1 of the Medicines Amendment Regulations 2009 (SR 2009/212 prescription, restricted, and pharmacy-only medicines). It is not legal to sell *M. speciosa* without a license, although it is not illegal to possess it.
Australia [156,157]	Restricted	Both mitragynine, one of the active chemicals in *M. speciosa*, and *M. speciosa* were placed in Schedule 9 of the Australian Standard for the Uniform Scheduling of Drugs and Poisons (SUSDP) in 2005.

## Data Availability

Not applicable.

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
