# Peer review of "Mitragyna Species as Pharmacological Agents: From Abuse to Promising Pharmaceutical Products"

_life, 2022, doi:10.3390/life12020193_

Round 1
Reviewer 1 Report
All my concerns have been well addressed, this manuscript can be published as it is.Reviewer 2 Report
The authors presented a well-written review on the highlights of the genus Mitragyna species. The paper can be accepted for publication.
Reviewer 3 Report
This review is well conducted and written and covered every aspect regarding the therapeutic application, phytochemistry, toxicology, Traditional Medicinal uses, native and historical background and botanical origins of Mitragyna Species
This manuscript is a resubmission of an earlier submission. The following is a list of the peer review reports and author responses from that submission.
Round 1
Reviewer 1 Report
-identical or very similar sentences are repeated in the manuscript (pg. 2 ln. 51-54 and ln. 85-88; ; pg. 5-6 ln. 172-178 and pg. 9 ln. 255-258)
-there is no need for Fig. 1
-antidiabetes???
-dan and research publications (pg. 6, ln. 182-183)???
-chapter 5. Phytochemistry of Genus Mitragyna should be placed before chapter 4. The Genus Mitragyna in Traditional Medicine and Ethnomedical Uses
Author Response
Dear reviewer
Thank you for your suggestions and corrections to improve the quality of our articles. Overall extensive editing of English language and style has been carried out using professional native English editing from MDPI.
-identical or very similar sentences are repeated in the manuscript (pg. 2 ln. 51-54 and ln. 85-88; ; pg. 5-6 ln. 172-178 and pg. 9 ln. 255-258)
Answer: has been revised and changed (red highlight) page 2 lines 88-89 while in lines 50-55 it remains.
-there is no need for Fig. 1
Answer: has been removed
-antidiabetes???
Answer: has been revised (red highlight) page 8 lines 206
-dan and research publications (pg. 6, ln. 182-183)???
Answer: has been revised (red highlight) page 8 lines 213
-chapter 5. Phytochemistry of Genus Mitragyna should be placed before chapter 4. The Genus Mitragyna in Traditional Medicine and Ethnomedical Uses
Answer: has been placed 4. Phytochemistry of the Genus Mitragyna (from page 5 lines 165 to page 8 lines198) and 5. The Genus Mitragyna in Traditional Medicine and Ethnomedical Uses (from page 8 lines 199 to page 9 lines 248
Reviewer 2 Report
In this paper, the authors aim at reviewing the pharmacological effects in humans of the plant M. speciosa. This would be a very welcome review given the implications for the populations where there is a traditional consumption or usage of this plant. However, the manuscript is not well put together. There are a number of spelling mistakes and/or misspellings throughout the manuscript and sometimes the sentences in the text seem somewhat disconnected, as if they are just put together without being careful with how the whole will read.
Some examples are as follows:
Line 65: It should be written “which” instead of “with”.
Lines 70, 71: The sentence “This plant has also been reported to be abused and traded online in powders and extracts under the name smoke shops [27] and the internet market of kratom [28].” is very confusing. What is meant by abused online? Smoke shops is a name for extracts? It needs to be rewritten.
Line 166: It should be “have the main stem cut out” instead of “are cut off the main stem”.
Line 201: The statement “varying processes between 5 months-10 years” is excessively vague, the way these processes are carried out should be explained.
Lines 206-207: In this sentence the verb that renders the action is missing.
Line 208: The number of the reference is missing.
Line 210-213: In these sentences it is stated that M. speciosa both reduces symptoms of withdrawal (from another drug?) and causes addiction. A more detailed explanation is necessary.
Line 284 The information on the type of extract should always be given, as it can affect the composition of the extract.
Line 287: motility is a movement, it is not a “part”. This sentence needs rewriting.
Line 298: What is meant by "potentially" in the sentence “the methanolic extract of kratom potentially has antioxidant and antimicrobial activities”? The extract either has or it has not activity.
Line 309: the sentence is grammatically incorrect, there is a verb missing related to the aftereffects.
Line 310-311: The meaning of CYP and of OTC is not indicated.
Lines 341-354: The last two sentences of these paragraph seem disconnected from the rest of the text.
Line 387: This reviewer finds this whole section very confusing. On one hand there is the problem of the structure of the sentences which renders them difficult to understand and, on the other hand, there are contradictory statements, like “The use of single M. speciosa leaves is known to be able to stop opioid addiction” and “dependence and side effects of M. speciosa leaves tea compared to illegal opioid groups”.
Line 421: it would be interesting, since the authors state that there are numerous preparations of M. speciosa, that the authors could supply information about the type of preparation that is more sought for in the online market.
Line 430: What do the authors mean by “Schedules of the United Nations Drug Conventions”? I believe that the word “schedules” is being misapplied.
Table 3: Head shops are not “intelligent shops”.
As I stated earlier, the above comments are just some examples of the problems that were found in the manuscript. I would like to stress that there are more of those and that is why an in-depth revision to the text is strongly advised, both in terms of English usage and with regard to the structure and organization of contents.
Furthermore, there are a lot of repetitions in the text, for example, the composition of the plant, and its effects, keeps being mentioned, statements that M. speciosa is better known as Kratom is repeated throughout the text numerous times. The manuscript, therefore, needs to be revised not only as regards of English usage but also in terms of producing a more coherent and organised text.
In addition, the authors claim to have described the pharmacological activity associated with this plant but there is no proof of pharmacological properties discussed in the text. There is a list of possible such effects repeated throughout the text but the works whose references are given are not fully explored as they should be in a review article. Also, there is a number of effects that keep being mentioned throughout the text, some positive some negative, without a clear distinction being made about them. The form in which the plant is sold and/or consumed is also not clear. In short, by the end of the manuscript there is not a clear picture about the properties of M. especiosa that could be used in pharmaceutics.
It is, therefore, the opinion of this reviewer that the manuscript needs to have extensive revising and editing before it could be considered for publication in Life.
Author Response
Dear reviewer
Thank you for your suggestions and corrections to improve the quality of our articles.
Some examples are as follows:
Line 65: It should be written “which” instead of “with”.
Answer: has been changed (red highlight page 2 line 65)
Lines 70, 71: The sentence “This plant has also been reported to be abused and traded online in powders and extracts under the name smoke shops [27] and the internet market of kratom [28].” is very confusing. What is meant by abused online? Smoke shops is a name for extracts? It needs to be rewritten.
Answer: has been revised and changed (red highlight) page 2 linea 70-71.
Line 166: It should be “have the main stem cut out” instead of “are cut off the main stem”.
Answer: has been changed (red highlight) page 5 lines 162-163.
Line 201: The statement “varying processes between 5 months-10 years” is excessively vague, the way these processes are carried out should be explained.
Answer: has been revised and changed (red highlight) page 8 lines 232-235
Lines 206-207: In this sentence the verb that renders the action is missing.
Answer: has been revised and changed (red highlight) page 8-9 lines 239-242
Line 208: The number of the reference is missing.
Answer: has been changed (red highlight) page 9 line 242
Line 210-213: In these sentences it is stated that M. speciosa both reduces symptoms of withdrawal (from another drug?) and causes addiction. A more detailed explanation is necessary.
Answer: has been revised and added detail explanation (red highlight) page 2 line 244-248.
Line 284 The information on the type of extract should always be given, as it can affect the composition of the extract.
Answer: has been revised and changed (red highlight) page 9 line 276-277
Line 287: motility is a movement, it is not a “part”. This sentence needs rewriting.
Answer: has been revised and changed (red highlight) page 9 line 279-280
Line 298: What is meant by "potentially" in the sentence “the methanolic extract of kratom potentially has antioxidant and antimicrobial activities”? The extract either has or it has not activity.
Answer: has been revised and removed (red highlight) page 9 line 286-291.
Line 309: the sentence is grammatically incorrect, there is a verb missing related to the aftereffects.
Answer: has been revised and changed (red highlight) page 10 line 300-301
Line 310-311: The meaning of CYP and of OTC is not indicated.
Answer: has been revised and changed (red highlight) page 10 line 302-303.
Lines 341-354: The last two sentences of these paragraph seem disconnected from the rest of the text.
Answer: has been revised and separated pharagraph pages 10-11 line 340-354
Line 387: This reviewer finds this whole section very confusing. On one hand there is the problem of the structure of the sentences which renders them difficult to understand and, on the other hand, there are contradictory statements, like “The use of single M. speciosa leaves is known to be able to stop opioid addiction” and “dependence and side effects of M. speciosa leaves tea compared to illegal opioid groups”.
Answer: has been revised and changed (red highlight) page 12 line 397-424
Line 421: it would be interesting, since the authors state that there are numerous preparations of M. speciosa, that the authors could supply information about the type of preparation that is more sought for in the online market.
Answer: has been revised and changed (red highlight) page 12 line 435-444
Line 430: What do the authors mean by “Schedules of the United Nations Drug Conventions”? I believe that the word “schedules” is being misapplied.
Answer: has been revised and changed (red highlight) page 13 line 450-453.
Table 3: Head shops are not “intelligent shops”.
Answer: has been revised and changed (red highlight) page 14 Table 3.
As I stated earlier, the above comments are just some examples of the problems that were found in the manuscript. I would like to stress that there are more of those and that is why an in-depth revision to the text is strongly advised, both in terms of English usage and with regard to the structure and organization of contents.
Furthermore, there are a lot of repetitions in the text, for example, the composition of the plant, and its effects, keeps being mentioned, statements that M. speciosa is better known as Kratom is repeated throughout the text numerous times. The manuscript, therefore, needs to be revised not only as regards of English usage but also in terms of producing a more coherent and organised text.
Answer:
- We already revised our manuscript (red highlight) and we also overall extensive editing of English language and style has been carried out using professional native English editing from MDPI.
- we have reduced overly repetitive words in the text
In addition, the authors claim to have described the pharmacological activity associated with this plant but there is no proof of pharmacological properties discussed in the text. There is a list of possible such effects repeated throughout the text but the works whose references are given are not fully explored as they should be in a review article. Also, there is a number of effects that keep being mentioned throughout the text, some positive some negative, without a clear distinction being made about them. The form in which the plant is sold and/or consumed is also not clear. In short, by the end of the manuscript there is not a clear picture about the properties of M. especiosa that could be used in pharmaceutics.
It is, therefore, the opinion of this reviewer that the manuscript needs to have extensive revising and editing before it could be considered for publication in Life.
Answer:
- We have added a brief explanation regarding the correlation of potential beneficial health effects based on case study reports with preclinical test results on experimental animals which are described on page 10 lines 305-310 and page 15 lines 529-537.
- The form in which the plant is sold/or consumed has been explained on page 15 lines 501-503.
Reviewer 3 Report
This manuscript provides a reasonably comprehensive and readable review of introducing of Mitragyna and its potential pharmacological uses. Besides, the toxicity, limitation and legal issue of using Mitragyna are well documented in this manuscript. Taken together, this review is well categorized and summarized, and also is very interesting and informative for the reader. This manuscript is recommended for publication after the following minor comments are addressed.
- In line 282, “anti-nociceptive” should be “antinociceptive” to make consist with above mentioned in this manuscript.
- In line 300, “Mitragynine” should be “mitragynine”.
- In the last paragraph of section 6, page 11, “ speciosa” should be italic.
- In line 363, the author should give a clear view what the tablet is and the weight.
- In line 368, the author should give more information about this case, like it’s one time use or long term use?
Author Response
Dear reviewer
Thank you for your suggestions and corrections to improve the quality of our articles. Overall extensive editing of English language and style has been carried out using professional native English editing from MDPI.
- In line 282, “anti-nociceptive” should be “antinociceptive” to make consist with above mentioned in this manuscript.
Anwser: anti-nociceptive has been changed “antinociceptive” (with red highlight) page 9 lines 275
- In line 300, “Mitragynine” should be “mitragynine”.
Answer: has been changed (with red highlight) page 10 lines 292
- In the last paragraph of section 6, page 11, “ speciosa” should be italic.
Answer: has been revised and changed (with red highlight) page 11 line 345-354.
- In line 363, the author should give a clear view what the tablet is and the weight.
Answer: has been revised and changed page 11 line 363-367.
Tabayali et al. (2018) described a kratom user who developed a brief prodromal acute sickness, accompanied by increased liver enzymes with cholestatic characteristics and jaundice. In this aspect, kratom use is a crucial contributor to the spread of the opioid crisis, as therapeutic benefits are gained at the expense of potentially fatal adverse effects [39].
- In line 368, the author should give more information about this case, like it’s one time use or long term use?
Answer: has been revised and changed page 11 line 374-376.
In another case, a 64-year-old man had recurrent seizures after consuming M. speciosa leaf tea mixed with Datura stramonium for 1 month due to chronic pain after post-colostomy surgery [132].
Round 2
Reviewer 1 Report
-in the Abstract (pg. 1, ln. 43) delete the part and ethnomedine, because it is the same term as traditional medicine
-in keywords (pg. 1, ln. 45) is the typewriting mistake in 7-hydroxymitragyna
-do not repeat the same word several times in one sentence (e.g. pg. 5, ln. 152-154)
-the same paragraph is repeated twice (pg. 5, ln. 166-185)
-the sentence (pg. 6, ln. 224-226) is unclear and grammatically incorrect, as well as mostly repeated in the next sentence. The same situation is with the sentence pg. 7, ln. 249-251
-In the manuscript still are repeated very similar sentences for several times
-always are used leaves and not the single leaf collected from plants, it should be corrected in the whole manuscript
-pg. 13, ln. 367-368, ln. 378-380
-In many cases it is better to divide a long sentence into two or more shorter ones, so that everything would be understandable
-pg. 16, ln.. 521-522
-pg. 17, ln 566 either simplicia?
-pg. 19, ln. 622 the part of the sentence plants of the genus M. speciosa is incorrect, M. speciosa is a species and Mitragyna is the genus
-pg. 21, ln. 704 the Mitragyna plant?
Author Response
Dear Reviewer
Once Again, thank you for your suggestion and comment to improve quality of our paper.
-in the Abstract (pg. 1, ln. 43) delete the part and ethnomedine, because it is the same term as traditional medicine
Answer: “and ethnomedine” has been remove as can be shown in pg 1, line 41-42 (blue highlight)
-in keywords (pg. 1, ln. 45) is the typewriting mistake in 7-hydroxymitragyna
Answer: has been revised (page 1 line 44, with blue highlight)
-do not repeat the same word several times in one sentence (e.g. pg. 5, ln. 152-154)
Answer: has been revised (page 4 line 145-146, with blue highlight)
-the same paragraph is repeated twice (pg. 5, ln. 166-185)
Answer: has been revised (page 5 line 157-164, with blue highlight)
-the sentence (pg. 6, ln. 224-226) is unclear and grammatically incorrect, as well as mostly repeated in the next sentence. The same situation is with the sentence pg. 7, ln. 249-251
Answer: in page 6 line 224-226, we already revised (unclear and grammatically incorrect) in page 5 line 168-185 with blue highlight; in page 7 line 249-251, also has been removed.
-In the manuscript still are repeated very similar sentences for several times
Answer: has been removed and revised
-always are used leaves and not the single leaf collected from plants, it should be corrected in the whole manuscript
-pg. 13, ln. 367-368, ln. 378-380
Answer: has been removed and revised (page 9 line 281 and page 10 line 296-297, with blue highlight)
-In many cases it is better to divide a long sentence into two or more shorter ones, so that everything would be understandable
Answer: we already divided some long sentence into two or more (blue highlight)
-pg. 16, ln.. 521-522
Answer: has been revised (page 13 line 451, with blue highlight)
-pg. 17, ln 566 either simplicia?
Answer: has been revised (page 13 line 475)
-pg. 19, ln. 622 the part of the sentence plants of the genus M. speciosa is incorrect, M. speciosa is a species and Mitragyna is the genus
Answer: has been revised (page 15 line 525, with blue highlight)
-pg. 21, ln. 704 the Mitragyna plant?
Answer: “the Mitragyna plant” has been replaced with “M. speciosa” (in page 17 line 609, with blue highlight)

Reviewer 2 Report
The authors have tried to comply with what was pointed out by the reviewer providing a revised manuscript that shows major improvements. However, there are still many issues that need to be addressed before the manuscript can be considered by this reviewer ready for publication. Some of these issues occur simply through lack of attention to detail. For example, in your reply, when you indicate the lines or pages where alterations were made, the numbers indicated were always wrong. This, obviously, made the process of reviewing more difficult and time-consuming than necessary. Also, despite the authors having had the English of the manuscript professionally revised, it seems that this was not done in the final version of the paper as there are paragraphs that are not syntactically or semantically correct. Finally, there are several parts where the text has problems, like repetitions and illegibility, which shows that a proper and careful revision was not made. Please consider the following comments:
M. speciosa is often written without being in italic. Please correct this simply by running a find and replace word search of the whole document.
Line 83: The meaning of the FDA initials was not given, as it should have been.
Lines 95, 314, 343, 446 and 595: Some sort of overlaying has occurred in these lines, rendering them impossible to read and, therefore, to assess the content.
Line 159: The alkaloid content that is being mentioned relates to which part of the plant? The leaves only? The question of the part of the plant that is being considered must be made clear right from the beginning because, depending on the plant, the compounds of interest may be in the leaf, flower, root or even stems. Therefore, the authors should make very clear which is the part of interest in this plant, and this section would be the most appropriate for this.
Lines 247-256: The beginning and the end of this paragraph need rewriting. The first sentence is syntactically incorrect and in the last sentence the word “conditions” does not seem to make much sense. Please rewrite/explain.
Lines 349-354: This paragraph needs rewriting as it does not make sense as it is.
Lines 378 – 380: Repetition of the same title.
Line 426: In what form is the plant mixed with the coffee? Whole leaves, powdered leaves, some sort of extract? Once again, it is not clear how, exactly, the plant is used.
Line 461: what is the origin of these compounds? And what do the authors mean with this information? The relevance and/or interpretation should be explained.
Line 470: Please specify how much is a “high dose” and how long is “a long time”.
Lines 473-475: Given that 79 mgx3= 237mg, where does the 214.29 mg come from? Please explain.
Line 480-487: There is still a lack of clarity with regard to the problem of the plant being able, on one hand, to relieve symptoms of addiction (from another drug) and on the other hand being able to cause addiction. In this paragraph, values are presented in a confusing way. The authors must be more systematic and separate what is related to “symptom relief”, from “cause of dependence” so that they can then perform a risk/benefit assessment on the usage of this plant. Besides, was the mitragynine that was administered, a pure compound isolated from M. speciosa? Or something else altogether? Once again there is a lack of detail and specifics that need to be provided. Furthermore, the authors are referring to drug administration in two different forms i.p. and s.c. (these initials need to be identified), which can, by themselves, impact on several parameters regarding the effect of a drug, toxicity being one of them. This issue, should, therefore, have been raised and discussed.
Lines 522 – 523 Repetition of the same title
Line 537: What exactly is meant by “powdered form”? What is the powder made of? The plant leaf?
Lines 543 to 558 should have been cut out as they are a repetition of the text that starts in line 560.
Line 549: What is meant by “simplicia”?
Line 596: This sentence requires better context. To what people are the authors referring to? In general, in a particular country…?
Line 648: “Despite” or “because”? The latter form would make more sense.
Line 661 should be deleted.
Line 681: what “green extraction method”? Needs clarification.
Author Response
Dear Reviewer
Once Again, thank you for your suggestion and comment to improve quality of our paper.
The authors have tried to comply with what was pointed out by the reviewer providing a revised manuscript that shows major improvements. However, there are still many issues that need to be addressed before the manuscript can be considered by this reviewer ready for publication. Some of these issues occur simply through lack of attention to detail. For example, in your reply, when you indicate the lines or pages where alterations were made, the numbers indicated were always wrong. This, obviously, made the process of reviewing more difficult and time-consuming than necessary. Also, despite the authors having had the English of the manuscript professionally revised, it seems that this was not done in the final version of the paper as there are paragraphs that are not syntactically or semantically correct. Finally, there are several parts where the text has problems, like repetitions and illegibility, which shows that a proper and careful revision was not made. Please consider the following comments:
- speciosais often written without being in italic. Please correct this simply by running a find and replace word search of the whole document.
Answer: we already revised (hopefully nothing has been missed)
Line 83: The meaning of the FDA initials was not given, as it should have been.
Answer: has been revised (page 2 line 77, with blue highlight)
Lines 95, 314, 343, 446 and 595: Some sort of overlaying has occurred in these lines, rendering them impossible to read and, therefore, to assess the content.
Answer: all of them, we already revised (blue highlight)
Line 159: The alkaloid content that is being mentioned relates to which part of the plant? The leaves only? The question of the part of the plant that is being considered must be made clear right from the beginning because, depending on the plant, the compounds of interest may be in the leaf, flower, root or even stems. Therefore, the authors should make very clear which is the part of interest in this plant, and this section would be the most appropriate for this.
Answer: has been revised (page 4 line 149-151, with blue highlight)
Lines 247-256: The beginning and the end of this paragraph need rewriting. The first sentence is syntactically incorrect and in the last sentence the word “conditions” does not seem to make much sense. Please rewrite/explain.
Answer: this paragraph we have already removed and we have combined some of the meanings in the sentence in the previous sentence to avoid repetitive (page 7 line 175-185), with blue highlight)
Lines 349-354: This paragraph needs rewriting as it does not make sense as it is.
Answer: has been revised (page 9 line 254-262)
Lines 378 – 380: Repetition of the same title.
Answer: has been revised (page 10, line 306)
Line 426: In what form is the plant mixed with the coffee? Whole leaves, powdered leaves, some sort of extract? Once again, it is not clear how, exactly, the plant is used.
Answer: has been revised in page 11 line 351-356 (blue highlight)
Line 461: what is the origin of these compounds? And what do the authors mean with this information? The relevance and/or interpretation should be explained.
Answer: has been removed and revised
Line 470: Please specify how much is a “high dose” and how long is “a long time”.
Answer: has been revised in page 11 line 390-392 (blue highlight)
Lines 473-475: Given that 79 mgx3= 237mg, where does the 214.29 mg come from? Please explain.
Answer: has been revised in page 12 line 396-403), there is a slight error in writing the mitragynine content number
Line 480-487: There is still a lack of clarity with regard to the problem of the plant being able, on one hand, to relieve symptoms of addiction (from another drug) and on the other hand being able to cause addiction. In this paragraph, values are presented in a confusing way. The authors must be more systematic and separate what is related to “symptom relief”, from “cause of dependence” so that they can then perform a risk/benefit assessment on the usage of this plant. Besides, was the mitragynine that was administered, a pure compound isolated from M. speciosa? Or something else altogether? Once again there is a lack of detail and specifics that need to be provided. Furthermore, the authors are referring to drug administration in two different forms i.p. and s.c. (these initials need to be identified), which can, by themselves, impact on several parameters regarding the effect of a drug, toxicity being one of them. This issue, should, therefore, have been raised and discussed.
Answer: has been revised in page 12 line 404-447, overall, we tried to explain separately between addiction effect and symptoms relief (as can been seen in page 11-12, line 385-447)
Lines 522 – 523 Repetition of the same title
Answer: has been revised (page 13 line 451 with blue highlight)
Line 537: What exactly is meant by “powdered form”? What is the powder made of? The plant leaf?
Answer: “powdered form” means the powder form of the dried leaf of M. speciosa, that has been revised in page 13 line 465-466, with blue highlight)
Lines 543 to 558 should have been cut out as they are a repetition of the text that starts in line 560.
Answer: has been removed (page 13 line 470-481)
Line 549: What is meant by “simplicia”?
Answer: “simplicial” means the dried leaf of M. speciosa, that has been revised in page 13 line 475-476
Line 596: This sentence requires better context. To what people are the authors referring to? In general, in a particular country…?
Answer: has been revised in page 14 line 499-500 (blue highlight)
Line 648: “Despite” or “because”? The latter form would make more sense.
Answer: has been replaced in page 16 line 551
Line 661 should be deleted.
Answer: has been removed
Line 681: what “green extraction method”? Needs clarification.
Answer: “green extraction method” means the utilization of nonconventional extraction method to obtain the optimum secondary metabolite target from natural product, that has been revised in page 16 line 583-585 (blue highlight)
